**Data Availability Statement:** All files are available from the PubMed database (DOIs: 10.1056/NEJMoa2108447, 10.1007/s00402-020-03461-z,

# Rerupture outcome of conservative versus open repair versus minimally invasive repair of acute Achilles tendon ruptures: A systematic review and meta-analysis

**Haidong Deng**[1], **Xin Cheng**[2], **Yi Yang**[1], **Fang Fang**[3], **Jialing He**[3], **Yixin Tian**[3], **Tiangui Li**[4], **Yangchun Xiao**[5], **Yuning Feng**[1], **Peng Wang**[3], **Weelic Chong**[6], **Yang Hai**[6], **Yu Zhang**[1,3]*

1 Department of Orthopedic, Affiliated Hospital of Chengdu University, Chengdu, Sichuan, China, 2 West China Hospital, Sichuan University, Chengdu, Sichuan, China, 3 Center for Evidence Based Medical, Affiliated Hospital of Chengdu University, Chengdu, Sichuan, China, 4 The First People's Hospital of Longquanyi District, Chengdu, Sichuan, China, 5 Department of Neurosurgery, Affiliated Hospital of Chengdu University, Chengdu, Sichuan, China, 6 Thomas Jefferson University, Philadelphia, Pennsylvania, United States of America

* tnt1057@outlook.com

## Abstract

### Objective

To compare the rerupture rate after conservative treatment, open repair, and minimally invasive surgery management of acute Achilles tendon ruptures.

### Design

Systematic review and network meta-analysis.

### Data sources

We searched Medline, Embase, and the Cochrane Central Register of Controlled Trials from inception to August 2022.

### Methods

Randomised controlled trials involving different treatments for Achilles tendon rupture were included. The primary outcome was rerupture. Bayesian network meta-analysis with random effects was used to assess pooled relative risks (RRs) and 95% confidence intervals. We evaluated the heterogeneity and publication bias.

### Results

Thirteen trials with 1465 patients were included. In direct comparison, there was no difference between open repair and minimally invasive surgery for rerupture rate (RR, 0.72, 95% CI 0.10–4.4; $I^2$ = 0%; Table 2). Compared to the conservative treatment, the RR was 0.27 (95% CI 0.10–0.62, $I^2$ = 0%) for open repair and 0.14 (95% CI 0.01–0.88, $I^2$ = 0%) for

10.1302/0301-620X.102B7.BJJ-2019-0783.R3, 10.1016/j.fas.2019.07.011, 10.1053/j.jfas.2019.02. 002, 10.1177/1071100718757971, 10.1177/ 0363546516651060, 10.1007/s00264-012-1737-9, 10.1007/s00590-013-1350-7, 10.1177/ 0363546513503282, 10.1302/0301-620X.93B8. 25998, 10.2106/JBJS.I.01401, 10.1177/ 0363546510376052).

**Funding:** The authors received no specific funding for this work.

**Competing interests:** The authors have declared that no competing interests exist.

minimally invasive surgery. The network meta-analysis had obtained the similar results as the direct comparison.

## Conclusion

Both open repair and minimally invasive surgery were associated with a significant reduction in rerupture rate compared with conservative management, but no difference in rerupture rate was found comparing open repair and minimally invasive surgery.

## Introduction

Although Achilles tendon is the strongest and thickest tendon, it is one of the most common tendon ruptures with an annual incidence of 37 to 50 per 100 000 persons, with the largest increase occurring in the middle-aged people [1–3].

Currently available treatments for Achilles tendon ruptures include conservative treatment and two types of surgical repair, open repair and minimally invasive surgery with the percutaneous and mini-open techniques [4, 5]. The risk of rerupture has been a major concern in the shared decisions making process between patient and physician. A recent meta-analysis showed that nonoperative treatment of Achilles tendon rupture has a higher risk of rerupture compared with operative treatment [6]. However, what types of surgical repair have a lower rerupture rate is still unclear.

Most recent systematic reviews comparing surgical techniques found that no relevant discrepancies were detected in terms of rerupture between open repair and minimally invasive surgery [7, 8]. Yet the majority of patients included in these meta-analyses were treated before 2010, with earlier generation devices. These meta-analyses may have represented earlier experience with Achilles tendon ruptures treatment. Since 2010, the percutaneous Achilles Repair System and several minimally invasive techniques have been described [5]. The development of surgical techniques and rehabilitation protocols in the last decade may contribute to lower odds of rerupture [9, 10].

Network Meta-Analyses is a quantitative data synthesis approach that enables both direct and indirect evaluation of multiple intervention models, hence providing more comprehensive insights into the clinical efficacy and acceptability of interventions [11–13]. We carried out this network meta-analysis to compare the rerupture rate after conservative treatment, open repair, and minimally invasive surgery of acute Achilles tendon ruptures.

## Methods

### Protocol and guidance

This systematic review and network meta-analysis was performed according to the Preferred Reporting Items for Systematic Reviews and Meta-Analyses (PRISMA) extension statement for reporting systematic reviews incorporating network meta-analysis [14]. The protocol was pre-registered in PROSPERO (CRD 42022340654).

### Data sources

We performed a comprehensive search to studies indexed until August 2022 in Medline, Embase, and the Cochrane Central Register of Controlled Trials from the electronic database.

We limited the search to humans, and all publications were made in English. The specifics of search terms were shown in the S1 Table.

### Eligibility criteria

We included all published RCTs when they met the following criteria:

Participants: (1) enrolled patients with acute Achilles tendon rupture; (2) patients were aged 16 years or older; (3) comparing different treatment options (conservative treatment, open repair treatment, or minimally invasive surgery); (4) within four weeks of rupture; (5) reported of re-rupture rate; (6) Studies published before 2010 were excluded.

Interventions: open repair, minimally invasive surgery

Comparison: Conservative

Outcomes: The primary outcome was rerupture rate. Secondary outcomes included wound infection, sural nerve injury, and deep vein thrombosis.

### Study selection

Two reviewers independently (XC and HD) selected studies by screening titles and abstracts and evaluating potential full-text. For research that has several publications, we included only the studies with the most informative and complete data. Discrepancies between reviewers were resolved by discussion or consulting a third author (YZ).

### Data collection process

Two reviewers (XC and HD) extracted data independently using a standardized form, including the type of study, intervention details and control characteristics, sample size, mean age, outcome measures, and follow-up intervals. A third reviewer examined the extracted data for mistakes. The consensus was reached during meetings.

### Risk of bias assessment

Two reviewers (XC and HD) independently assessed the risk of bias of included trials using the Cochrane Risk of Bias tool across seven domains [15, 16]. Each trial received a study-level score of low, high, or unclear risk of bias for each domain. Discrepancies were resolved by consensus, and a third author (YZ) gave a final judgment if no consensus was achieved.

### Data synthesis

This network meta-analysis was performed by using R software (version 4.1.0) with the package gemtc (version 1.0–1) that based on Bayesian framework. R software interfaces with JAGS software (version 4.3.0) were applied to computing Markov chain Monte Carlo operation to conduct a multiple treatments comparison [17].

For pairwise meta-analysis, we used a random effects model to compute pooled effect sizes, and risk ratio for outcomes with 95% confidence intervals.

For network meta-analysis, we used a random consistency model to compute the study effect sizes, and binomial likelihood arguments for the rerupture outcome [18]. Treatment effects were estimated using risk ratio (RR) for dichotomous outcomes with 95% confidence intervals [19].

Specifically, we established 4 independent Markov chains with over dispersed initial values, 50,000 simulations for each chain were discarded as burn-in period. Then, 100,000 sample

iterations per chain simultaneously to ensure model convergence. The Brooks-Gelman-Rubin plots approach was used to evaluate model convergence, with the potential scale reduction factor (PSRF) serving as the assessment indicator [20]. PSRF values close to one indicate the complete convergence effect of the model.

We used node-splitting analysis to determine the inconsistency of the model between direct and indirect comparisons [21]. P-value less than 0.05 suggests the consistency of the model is satisfactory.

We also assessed the global heterogeneity, using the anohe function of the 'gemtc' package to calculate the bias of the magnitude of heterogeneity variance parameter $I^2$.

To assess the transitivity assumption, we examined the distribution of clinical and methodological characteristics. (e.g., age, sex, treatment protocol).

## Results

### Characteristics of included studies

Fig 1 shows the specific study screening flowchart. A total of 737 citations were identified from the databases. After removing duplicates and screening the title and abstract, 33 studies were selected for a full-text review. Finally, 13 trials met the inclusion eligibility criteria [22–34]. The included studies were published between 2010 and 2022. The length of time post-operative that the most studies we included was about 12 to 24 months.

Fig 2 presents the network plot. The 13 studies included 1465 patients, of whom 500 were conservative treatment, 583 were treated with open repair, and 340 were treated with minimally invasive surgery. The characteristics of included randomized controlled trials were displayed in Table 1.

### Risk of bias in included studies

S6 and S7 Figs showed the risk-of-bias assessments. Five trials were low risk of bias, four trials were unclear risk, and four trials were high risk. The primary bias was the blinding of outcome assessment.

### Rerupture

The pair-wise meta-analysis pooled effects showed that no difference between open repair and minimally invasive surgery for rerupture rate (RR, 0.72, 95% CI 0.10–4.4; $I^2$ = 0%; Table 2). Compared to the conservative treatment, the RR was 0.27 (95% CI 0.10–0.62, $I^2$ = 0%) for open repair and 0.14 (95% CI 0.01–0.88, $I^2$ = 0%) for minimally invasive surgery. The network meta-analysis had obtained the similar results as the direct comparison (S1 Fig). No statistically significant differences were found in rerupture rate between open repair and minimally invasive surgery.

### Other outcomes

Compared to conservative treatment, open repair management had significant higher infection rate, with very wide confidence intervals (S3 Fig). There was a significant difference between conservative treatment and open repair management in deep vein thrombosis complication (S4 Fig). We have not found any significant difference in sural nerve injury (S5 Fig).

### Model fit and evaluation of consistency

The PSRF value was 1.000, indicating a strong iterative effect, complete convergence, and stable model outputs (S2 Fig).

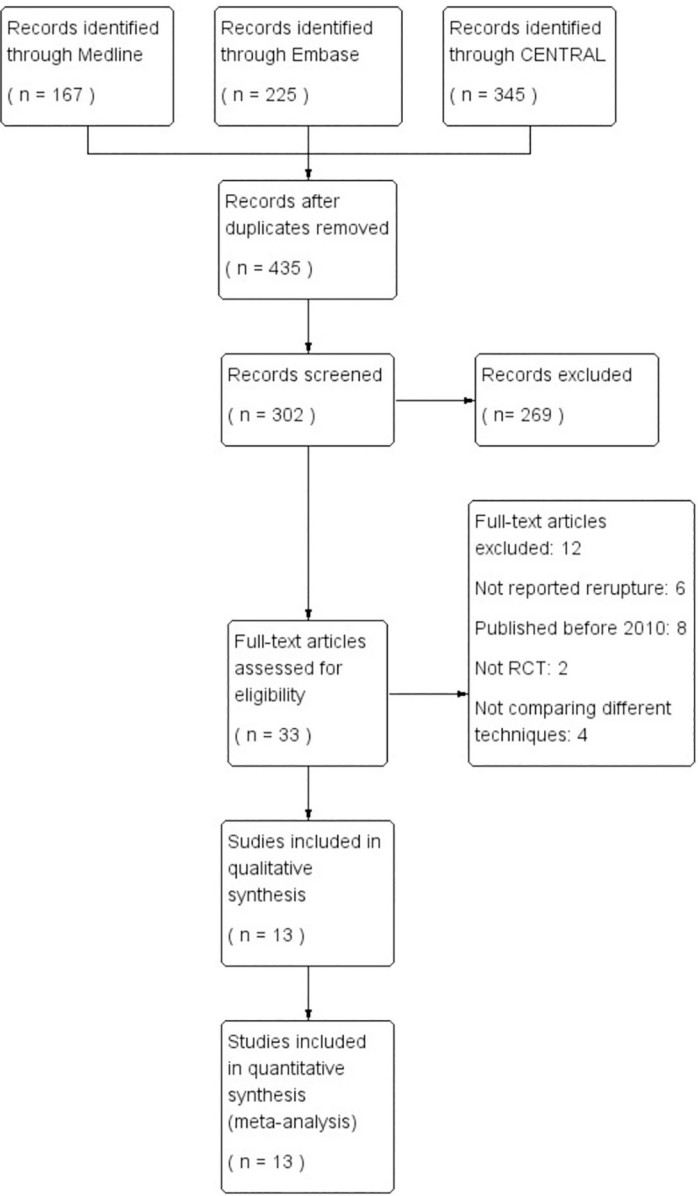

**Fig 1. Search strategy and final included and excluded studies.**

The node-splitting analysis showed that there was no inconsistency between direct and indirect comparisons among conservative, open repair, and minimally invasive surgery with P > 0.05 (Fig 3). There was also no existence of heterogeneity in the direct and indirect comparisons (S2 Table). The global I- squared was 0.

## Discussion

### Principal findings

This systematic review and network meta-analysis of RCTs performed a comparison among conservative treatment versus open repair versus minimally invasive surgery for acute Achilles tendon ruptures. The mixed results showed that both open repair and minimally invasive

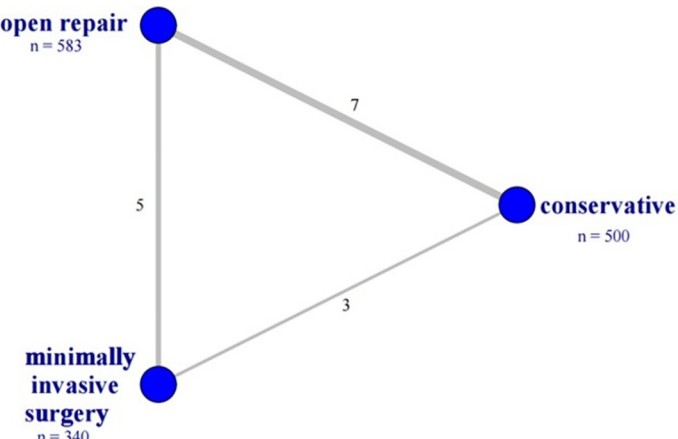

**Fig 2. Network plot of the direct comparisons of the re-rupture for all included studies.** The network geometry for risk of rerupture. The number of participants is showed by n = below the treatment name. Width of the lines is proportional to the number of trials informing an indicated comparison and is specified with the number adjacent the edge.

surgery were associated with a significant reduction in rerupture rate compared with conservative management, but no difference in rerupture rate was found comparing open repair and minimally invasive surgery.

## Comparison with previous findings

The previous study has demonstrated that operative treatment (open repair and minimally invasive surgery) of acute Achilles tendon ruptures could reduce the risk of re-rupture compared with nonoperative treatment [6]. Still, which types of surgical repair have a lower rerupture rate is unknown. Meulenkamp et al. [35] performed an network meta-analysis based on the best available evidence to guide the management of acute achilles tendon ruptures. The risk of rerupture outcome showed that there was no difference in rerupture risk between open surgical repair, minimally invasive surgery repair, and functional rehabilitation. Yet, primary immobilization was associated with a higher rerupture risk than open repair. However, they did not include the current largest RCT that compared nonoperative treatment, open repair, and minimally invasive surgery in patients with acute Achilles tendon rupture. More recently, Gatz et al. [8] conducted a meta-analysis that included RCTs and observational studies, including 25 studies with a total of 2223 patients, and Attia et al. [7] performed a meta-analysis included RCTs only, including 10 trials with a total of 522 patients. Both studies compared the rerupture rate between open repair and minimally invasive surgery of acute Achilles tendon ruptures. The pooled effect showed no relevant differences in re-rupture rate between the two techniques.

Our review showed differences in design and settings. Firstly, this network meta-analysis used direct and indirect method to compare the risk of rerupture between conservative treatment, open repair, and minimally invasive surgery for acute Achilles tendon ruptures. Secondly, we excluded studies published before 2010 to eliminate the impact of early techniques and rehabilitation protocols. Thirdly, we only included RCTs to remove the inherent selection bias.

## Limitations

As one of the primary causes of Achilles tendon surgical failure, we solely investigated the rate of re-rupture. We did not analyze other complications such as pulmonary embolism, deep

**Table 1. Characteristics of included randomized controlled trials.**

| Studies | Number | | | Sex (femle/male) | | | Mean (SD or range) age, years | Follow-up (months) | Treatment Arms |
|---|---|---|---|---|---|---|---|---|---|
| | conservative | Open repair | minimally invasive surgery | conservative | Open repair | minimally invasive surgery | | | |
| Nilsson-Helander 2010 | 48 | 49 | - | 9/39 | 9/40 | - | 41.2 (23–63); 40.9 (24–59) | 12 | plaster cast for 2 wk + orthosis for 6 wk (WB at wk 6–8); OR + plaster cast for 2 wk + orthosis for 6 wk (WB at wk 6–8) |
| Willits 2010 | 72 | 72 | - | 13/59 | 13/59 | - | 41.1 (8.0); 39.7 (11.0) | 24 | orthosis for 8 wk (WB at wk 2); OR + splint for 2 wk + orthosis for 8 wk |
| Keating 2011 | 41 | 39 | - | 9/32 | 11/28 | - | 39.5 (21–58); 41.2 (27–59) | 12 | plaster cast for 10 wk (WB at wk 8); Open repair + plaster cast for 6 wk |
| Olsson 2013 | 51 | 49 | - | 4/47 | 10/39 | - | 39.5 (9.7); 39.8 (8.9) | 12 | orthosis for 8 wk (WB immediately); OR + orthosis for 6 wk (WB immediately) |
| Kararinas 2014 | - | 15 | 19 | - | 2/13 | 4/15 | 40 (28–50); 42 (25–58) | 24 | OR + plaster cast for 3 wk + orthosis for 3–4 wk; MIS + plaster cast for 3 wk + orthosis for 3–4 wk |
| Kolodziej 2013 | - | 27 | 24 | - | 1/26 | 1/23 | 47.1 (26–74); 44.8 (30–62) | 24 | OR + plaster cast for 6 wk (WB at 6 wk); MIS + plaster cast for 6 wk (WB at 6 wk) |
| Lantto 2016 | 28 | 32 | - | 3/25 | 2/30 | - | 39 (28–60); 40 (27–57) | 18 | plaster cast for 1 wk + orthosis for 6 wk; OR + plaster cast for 1 wk + orthosis for 6 wk |
| Rozis 2018 | - | 41 | 41 | - | 9/32 | 1031 | 41 (19.5); 43 (18.5) | 12 | OR + plaster cast for 3 wk + orthosis for 5 wk (WB at 4); MIS + plaster cast for 3 wk + orthosis for 5 wk (WB at 4) |
| Manent 2019 | 11 | 12 | 11 | 1/10 | 1/11 | 1/10 | 42 (26–51); 40.5 (28–51); 41 (18–50)* | 12 | OR + plaster cast for 1.5 wk + orthosis for 4.5 wk; MIS + plaster cast for 1.5 wk + orthosis for 4.5 wk; plaster cast |
| Makulavicius 2020 | - | 44 | 43 | - | 5/39 | 5/38 | 37.8 (10.1); 35.9 (9.5) | 36 | OR + plaster cast for 3 wk + orthosis for 2–3 wk; MIS + plaster cast for 3 wk + orthosis for 2–3 wk |
| Maempel 2020 | 41 | 39 | - | 9/32 | 11/28 | - | 39.5 (21–58); 41.2 (27–59) | 188 | plaster cast for 10 wk (WB at wk 8); Open repair + plaster cast for 6 wk |
| Fischer 2021 | 30 | 30 | 30 | 3/27 | 2/28 | 4/26 | 45.2 (9.5); 39.3 (7.9); 39.6 (7.3) | 24 | plaster cast for 6 wk + orthosis for 2 wk (WB immediately); OR + plaster cast for 6 wk + orthosis for 2 wk (WB immediately); MIS + plaster cast for 6 wk + orthosis for 2 wk (WB immediately) |
| Myhrvold 2022 | 178 | 176 | 172 | 42/136 | 44/132 | 49/123 | 39.9 (8.1); 39.9 (8.9); 39.1 (8.4) | 12 | plaster cast for 2 wk + orthosis for 6 wk (WB at 2 wk); OR + plaster cast for 2 wk + orthosis for 6 wk (WB at 2 wk); MIS + plaster cast for 2 wk + orthosis for 6 wk (WB at 2 wk) |

* Data are shown as median (range)

WK: week; OR: open repair; MIS: minimally invasive surgery; WB: weightbearing.

**Table 2. Comparison of different treatment in Achilles tendon rupture between pair-wise meta-analysis and network meta-analysis.**

|  | Conservative treatment vs open repair | Conservative treatment vs minimally invasive surgery | Open repair vs minimally invasive surgery |
|---|---|---|---|
| pair-wise meta-analysis | 0.27 (0.11, 0.59) | 0.17 (0.04, 0.71) | 0.62 (0.14, 2.63) |
| network meta-analysis | 0.27 (0.10, 0.62) | 0.14 (0.01, 0.88) | 0.72 (0.10, 4.4) |

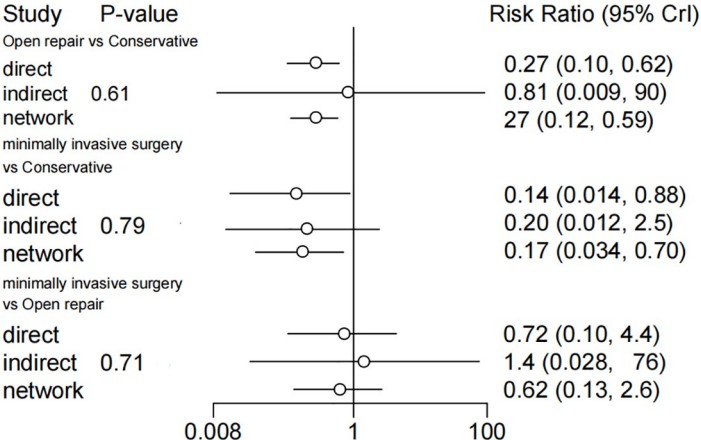

**Fig 3. Node-splitting analysis of inconsistency for rerupture rate.** Summary of a node-splitting analysis consisting of separate node-splitting models and a consistency model.

vein thrombosis, sural nerve injury, wound infection, etc. We failed to obtain appropriate date to analyze the outcome of long-term rerupture rate due to those majority patients were followed up to 12 to 24 months. Additionally, we conducted a sensitive analysis to excluded the influence of the study that had 188 months follow-up. Through analysis, we obtained a similar result. We failed to examine the complication of Pulmonary embolism due to only two studies report it in this paper. The results of the network meta-analysis for indirect comparison, however, were met with low to very low levels of confidence, mostly because of imprecision around the effect estimates and within-study bias. The relative treatment estimates may vary as a result of future high-quality research, even if these first results are optimistic due to the low certainty of the evidence for rerupture.

## Conclusion

The results of the present study suggest that both open repair and minimally invasive surgery were associated with a significant reduction in rerupture rate compared with conservative management, but no difference in rerupture rate was found comparing open repair and minimally invasive surgery.

## Supporting information

**S1 Table. Search strategy.**
(DOCX)

**S2 Table. Analysis of heterogeneity for rerupture rate.** T1: treatment 1; T2: treatment 2; $I^2$. pair: I-square of pair-wise meta-analysis; $I^2$.cons: I-square of network meta-analysis; Incons.p: inconsistency p-values for pair-wise and network meta-analysis.
(DOCX)

**S1 Fig. Forest plot of re-rupture rate in network meta-analysis of achilles tendon ruptures.**
(TIF)

**S2 Fig. Brooks–Gelman–Rubin plots.**
(TIF)

**S3 Fig. Node-splitting analysis of inconsistency for the wound infection.**
(TIF)

**S4 Fig. Node-splitting analysis of inconsistency for the deep vein thrombosis.**
(TIF)

**S5 Fig. Node-splitting analysis of inconsistency for the sural nerve injury.**
(TIF)

**S6 Fig. Risk of bias summary.**
(TIF)

**S7 Fig. Risk of bias graph.**
(TIF)

**S1 Checklist. PRISMA NMA checklist of items to include when reporting a systematic review involving a network meta-analysis.**
(DOCX)

**S2 Checklist. PRISMA 2020 checklist.**
(DOCX)

## Author Contributions

**Conceptualization:** Haidong Deng, Fang Fang, Yixin Tian, Tiangui Li, Yangchun Xiao, Yuning Feng, Peng Wang, Weelic Chong, Yang Hai, Yu Zhang.

**Data curation:** Haidong Deng, Yi Yang, Tiangui Li.

**Formal analysis:** Haidong Deng, Xin Cheng, Yi Yang.

**Investigation:** Yi Yang, Yuning Feng.

**Methodology:** Haidong Deng, Xin Cheng, Tiangui Li, Yuning Feng, Yu Zhang.

**Project administration:** Fang Fang, Yu Zhang.

**Software:** Haidong Deng, Xin Cheng, Yi Yang, Yixin Tian, Yangchun Xiao.

**Supervision:** Yu Zhang.

**Validation:** Jialing He, Yixin Tian, Peng Wang, Yang Hai.

**Visualization:** Fang Fang, Jialing He, Peng Wang.

**Writing – original draft:** Haidong Deng, Yu Zhang.

**Writing – review & editing:** Haidong Deng, Xin Cheng, Yi Yang, Fang Fang, Jialing He, Yixin Tian, Yangchun Xiao, Yuning Feng, Peng Wang, Weelic Chong, Yang Hai, Yu Zhang.

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
