## [Decision Letter · Decision Letter 0]

20 Feb 2023

PONE-D-22-33292

Comparative relative effects between conservative, open repair and minimally invasive surgery treatment for Achilles tendon rupture: a systematic review and network meta-analysis

PLOS ONE

Dear Dr.Zhang

Thank you for submitting your manuscript to PLOS ONE. After careful consideration, we feel that it has merit but does not fully meet PLOS ONE’s publication criteria as it currently stands. Therefore, we invite you to submit a revised version of the manuscript that addresses the points raised during the review process.

Thank you for submitting your manuscript to PLOS ONE. Three reviewers have completed external review of the manuscript. Due to the differences in decision between the reviewers, I would advice you to carefully consider all comments from reviewer 2 and 3. 

We look forward to receiving your revised manuscript.

Kind regards,

Daniel Ramskov, Ph.D

Academic Editor

PLOS ONE

Journal Requirements:

“No, the funders had no role in study design, data collection and analysis, decision to publish, or preparation of the manuscript.”

Reviewers' comments:

Reviewer's Responses to Questions

**Comments to the Author**

1. Is the manuscript technically sound, and do the data support the conclusions?

Reviewer #1: Yes

Reviewer #2: Yes

Reviewer #3: Yes

2. Has the statistical analysis been performed appropriately and rigorously? 

Reviewer #1: Yes

Reviewer #2: Yes

Reviewer #3: Yes

3. Have the authors made all data underlying the findings in their manuscript fully available?

Reviewer #1: Yes

Reviewer #2: Yes

Reviewer #3: Yes

4. Is the manuscript presented in an intelligible fashion and written in standard English?

Reviewer #1: Yes

Reviewer #2: Yes

Reviewer #3: No

5. Review Comments to the Author

Reviewer #1: This is a very simple network meta-analysis with only 3 comparisons, two of which were indirect (MIS vs. conservative and open repair vs. conservative) and one direct comparison for the two interventions of interest (MIS vs. open repair). The assumption being that there were no appreciable difference in the interventions be applied. The authors note this early on in their presentation. A risk of bias assessment was performed for the studies of interest as was a test of consistency for the direct vs. the indirect comparisons. There appeared to be little or no heterogeneity issues. Apparently there were 13 studies included in the final quantitative assessments. The overall sample size appears to be adequate.

The authors give an objective assessment of a major limitation of the study which was that the results of the network meta-analysis for indirect comparison, however, were met with low to very low levels of confidence, mostly because of imprecision around the effect estimates and within-study bias.

Since there were only two major interventions of interest, the staistical analysis as seen in the tables and figures, was fairly simple and the results are consistent with their assessment that both open repair and MIS were associated with a significant reduction in rerupture rate independently compared with conservative management, but there was no difference between open repair and MIS.

On a minor note, there were a few grammatical corrections needed.

Reviewer #2: I would like to congratulate the investigators in completing this meta analysis. Achilles Tendon rupture (ATR) is a common injury that could result in persisting symptoms and physical limitations. In the shared decision-making process between patient and physician about choosing treatment, knowledge of the risk of complications such as re-rupture is of major importance for treatment recommendations. We now know that on group-level conservative treatment (CT), open repair (OR), and minimally invasive surgery (MIS) yields similar patient reported outcomes, physical performance, and total risk of major complications (Myhrvold et al., 2022). However, rerupture remains the major risk factor for patients treated non-operatively and this complication has the largest impact on the acute Achilles tendon Total Rupture score (Metz et al., 2011).

Here is some issues that the authors may wish to address:

1. The title is a little confusing. The meta analysis compares of the risk of re-rupture between conservative treatment, open repair surgery, and minimally invasive surgery. The results are based on pairwise comparisons. I would suggest changing the title to make it more informative and easy to the reader.

2. Abstract Line 39. In the Objective section the authors state that they wanted "to compare the relative effects of rerupture rate...". Is the "relative effect" of re-rupture rate after a certain treatment really addressed in this paper? I suggest changing the first sentence in line 39-41 to simply: To compare the rerupture rate...

3. Abstract Line 46: Articles publish before 2010 was excluded. I suggest changing the word "inception" to "2010". Could the authors also explain why this year was chosen for including studies in the meta analysis? The first RCT comparing conservative treatment with surgery for ATR was published in 1981 by Lars Nistor. Since then, many RCTs comparing different treatments have been published. However, very few studies have compared the results after CT, OR, and MIS (3-armed studies). In the literature, there are very few publications of comparative 3-armed studies regarding treatment for ATR before 2010 (Schroeder et al., Treatment of acute Achilles Tendon ruptures: open vs. percutaneous repair vs. conservative treatment: a prospective randomized study. Orthop trans 1997;21:1228. And Majewski et al., Achilles Tendon rupture: a prospective study assessing various treatment possibilities. Orthopade 2000;29:670-6). Some readers might miss those studies in the analyses... Your choice.

4. Abstract Line 51: I would suggest using "confidence intervals" instead of "credible intervals".

5. Abstract Line 53: Suggest never to start a sentence with a number. Spell 13 out (Thirteen...).

6. Abstract Line 53-58: The information given here could be re-written and compressed giving the same information with fewer words.

7. Abstract Line 62: Please state what difference you refer to (re-rupture).

8. Introduction Line 68: Some studies report higher incidences than 31/100.000. Houshian et al. (1996) found an incidence of 37/100.000, and Huttunen et al. found an incidence close to 50/100.000 (Huttunen TT, Kannus P, Rolf C, Felländer-Tsai L, Mattila VM. Acute achilles tendon ruptures: incidence of injury and surgery in Sweden between 2001 and 2012. Am J Sports Med. 2014 Oct;42(10):2419-23. doi: 10.1177/0363546514540599. Epub 2014 Jul 23. PMID: 25056989.)

9. Line 71-72: When claiming there are "two types of surgical repair" the reader might wonder if the authors have forgotten about the percutaneous technique (described in ref. 4 and 5). Suggest changing the sentence a bit to avoid this potensial confusion.

10. Line 72-73: Suggest changing focus a bit. For Achilles´ tendon rupture: isn't it the patient who choses his/her treatment based on information given by the physician/surgeon of risks of complications and expected treatment results? And not the opposite? Suggest "The risk of rerupture is a major concern in the shared decision making process (between patient and physician)" or something like that.

11. Line 74: Again: It is not the surgeon, but the patient who choses the treatment, agree?. Furthermore, the trend in treatment of ATR is different between regions of the world: In Asia, surgery is more common than in Scandinavian countries for example. Remember that the literature is shared world wide. Despite that a large number of surgeons in Asia recommend surgical treatment to their patients, there are very few who do so in Scandinavia for instance.

(Yamaguchi S, Kimura S, Akagi R, Yoshimura K, Kawasaki Y, Shiko Y, Sasho T, Ohtori S. Increase in Achilles Tendon Rupture Surgery in Japan: Results From a Nationwide Health Care Database. Orthop J Sports Med. 2021 Oct 21;9(10):23259671211034128. doi: 10.1177/23259671211034128. PMID: 34708136; PMCID: PMC8543583).

(Leino O, Keskinen H, Laaksonen I, Mäkelä K, Löyttyniemi E, Ekman E. Incidence and Treatment Trends of Achilles Tendon Ruptures in Finland: A Nationwide Study. Orthop J Sports Med. 2022 Nov 8;10(11):23259671221131536. doi: 10.1177/23259671221131536. PMID: 36389616; PMCID: PMC9647260.)

(Wilder JH, Ofa SA, Lee OC, Gadinsky NE, Rodriguez RF, Sherman WF. Rates of Operative Management for Achilles Tendon Rupture Over the Last Decade and the Influence of Gender and Age. Foot Ankle Spec. 2022 Jun 13:19386400221102745. doi: 10.1177/19386400221102745. Epub ahead of print. PMID: 35695472.)

12. Line 82-83: Regarding change in the risk of re-rupture over time (before or after 2010). It has been claimed that accelerated rehabilitation lowers the risk of re-rupture after conservative treatment. Twaddle & Poon (Early motion for Achilles Tendon ruptures: is surgery important? A randomized prospective study.) introduced this theory in 2007 and was later supported by Soroceanu (Surgical versus non-surgical treatment of acute Achilles tendon rupture: a metaanalysis of randomized trials. J Bone Joint Surg Am 2012;94:2136-43). But it was a secondary finding and therefore only speculative. Later, two prospective cohort studies by Hutchison and Aujla investigated specific accelerated rehabilitation protocols in non-controlled cohorts confirming the low risk of re-rupture. However, the studies were suffering from an insufficient follow up leaving a high risk of patients with rerupture not being registered in the cohorts.

Suggest adding a reference in Line 83 and/or remove "..and rehabilitation protocols.."

(Hutchison AM, Topliss C, Beard D, Evans RM, Williams P. The treatment of a rupture of the Achilles tendon using a dedicated management programme. Bone Joint J. 2015 Apr;97-B(4):510-5. doi: 10.1302/0301-620X.97B4.35314. PMID: 25820890.)

(Aujla RS, Patel S, Jones A, Bhatia M. Non-operative functional treatment for acute Achilles tendon ruptures: The Leicester Achilles Management Protocol (LAMP). Injury. 2019 Apr;50(4):995-999. doi: 10.1016/j.injury.2019.03.007. Epub 2019 Mar 11. PMID: 30898390.)

13. Line 87: Suggest removing "relative effects of".

14. Line 106-108: The authors claim that the interventions were OR and MIS compared to CT. But in the Results section they report pairwise comparisons. Suggest pairwise comparisons also in the Methods section.

15. Line 114: Change "will be" to "were".

16. Line 132-133: Grammatically error.

17. Line 142: Remove "was".

18. Line 163-173: This section regarding reruptures can be shortened without losing information.

19. Line 184: Please be consistent. Chose "non-operative" or "conservative" throughout the paper. Also specify both OR and MIS.

20. Line 187: Same as point 7.

21. Line 189: operative treatment (open repair and minimally invasive surgery)

22. Line 201: Change "them" to "studies"

23. Line 202: Use either abbreviations on both open repair and minimally invasive surgery, or spell out for both, and be consistent throughout the whole manuscript.

24. Line 203-204. The language can be shortened throughout the manuscript. Here is an example of how the information could be presented: "The pooled effect showed no relevant differences in re-rupture rate between the two techniques."

25. Line 207: abbreviations or spell out.

26. Line 223-225: The conclusion is well written, but consider removing "independently" in line 224 and state "no difference in rerupture rate was found comparing open repair and minimally invasive surgery".

27. Line 403: Table 1 is Characteristics of included randomized controlled trials (spell out). Not Baseline characteristics.

28. Table 1: In the last study (Myhrvold 2022): add (WB at 2 wk) also after non-operative treatment.

29. Table 2 Line 406-412: Please do not use abbreviations in table and figure descriptions. Spell out. In my opinion table 2 is to difficult to grasp for the reader. Consider to present the pairwise comparisons more clearly with three columns but with the upper row stating Conservative treatment vs open repair; Conservative treatment vs minimally invasive surgery; Open repair vs minimally invasive surgery.

In conclusion I find the result of this study very interesting and important. In my opinion, however, prior to publication, the manuscript needs to be revised. I wish the authors well in further preparation.

Reviewer #3: The authors present an interesting study comparing the relative effects between conservative, open, and minimally invasive Achilles tendon repair for Achilles tendon rupture. They demonstrated that no differences exist in re-rupture rate between the open and minimally invasive techniques; however, but of these techniques had less re-rupture rates compared to conservative treatment.

Several questions arise regarding this study. What was the length of time post-operative that these patients were followed. Several studies have shown that operative management of Achilles tendon rupture have improvement in early re-rupture, but what about long term re-rupture rates between these techniques? (PMID: 7551757)

Unfortunately, as stated by the authors, several systematic reviews comparing these operative techniques have been performed and at a much more rigorous level. I fail to see how this study contributes anything new to the field of Achilles tendon rupture. What separates this study from those other studies? Why were other complications like sural nerve injury, wound dehiscence, VTE, surgical site infections not examined? Inclusion of these data would greatly improve your data.

The configuration of Table 2, which is the crux of your data, is extremely difficult to interpret. Re-configuration of this data would greatly improve the main message of this study.

As the main analysis was performed by XC and HD with YZ as the tie breaker, further explanation of the credentials of these reviewers is paramount to state to the reader. What qualifies these reviewers to understand, analyze, and interpret all this data?

This was a great effort by the authors to help determine best course of action for a growing problem especially in the post-COVID era where Achilles tendon ruptures are rampant as people are slowly returning to their normal athletic activities after several years without normal athletic activity patterns. Answering the above questions will greatly assist in the publication of this study.

6. PLOS authors have the option to publish the peer review history of their article (what does this mean?). If published, this will include your full peer review and any attached files.

Reviewer #1: No

Reviewer #2: **Yes: **Ståle Bergman Myhrvold

Reviewer #3: No

---

## [Author Response · Author response to Decision Letter 0]

28 Mar 2023

Re: Revision Manuscript ID PONE-D-22-33292, entitled “Rerupture outcome of Conservative Versus Open repair Versus Minimally Invasive Repair of Acute Achilles Tendon Ruptures: A Systematic Review and Meta-analysis"

March 16 2023 

Dear Dr. Emily Chenette, Dr. Daniel Ramskov, and Editorial Team: 

Thank you very much for the reviews of our manuscript. We have revised the manuscript using a word processing program and uploaded it as the main document. We have provided a point-by-point response to each of the editors’ and reviewers’ comments below in italicized blue text. 

Sincerely, 

Yu Zhang, on behalf of the authors.

Yu Zhang, MD, PhD

Affiliated Hospital of Chengdu University, Chengdu, Sichuan, China.

Chengdu, Sichuan, 610041, China.

E-mail address: zhangyu1057@cdu.edu.cn

 

 Reviewer #1: This is a very simple network meta-analysis with only 3 comparisons, two of which were indirect (MIS vs. conservative and open repair vs. conservative) and one direct comparison for the two interventions of interest (MIS vs. open repair). The assumption being that there were no appreciable difference in the interventions be applied. The authors note this early on in their presentation. A risk of bias assessment was performed for the studies of interest as was a test of consistency for the direct vs. the indirect comparisons. There appeared to be little or no heterogeneity issues. Apparently there were 13 studies included in the final quantitative assessments. The overall sample size appears to be adequate.

Authors’ Response: Thanks for your positive comments.

The authors give an objective assessment of a major limitation of the study which was that the results of the network meta-analysis for indirect comparison, however, were met with low to very low levels of confidence, mostly because of imprecision around the effect estimates and within-study bias.

Authors’ Response: Thanks for your positive comments.

Since there were only two major interventions of interest, the staistical analysis as seen in the tables and figures, was fairly simple and the results are consistent with their assessment that both open repair and MIS were associated with a significant reduction in rerupture rate independently compared with conservative management, but there was no difference between open repair and MIS.

Authors’ Response: Thanks for your positive comments.

On a minor note, there were a few grammatical corrections needed.

Authors’ Response: Thanks for your suggestion. We have changed the sentence. We tried our best to improve the manuscript and made some changes to the manuscript. These changes will not influence the content and framework of the paper. And here we did not list the changes but marked in red in the revised paper. We appreciate for Editors/Reviewers’ warm work earnestly and hope that the correction will meet with approval.

Reviewer #2: I would like to congratulate the investigators in completing this meta analysis. Achilles Tendon rupture (ATR) is a common injury that could result in persisting symptoms and physical limitations. In the shared decision-making process between patient and physician about choosing treatment, knowledge of the risk of complications such as re-rupture is of major importance for treatment recommendations. We now know that on group-level conservative treatment (CT), open repair (OR), and minimally invasive surgery (MIS) yields similar patient reported outcomes, physical performance, and total risk of major complications (Myhrvold et al., 2022). However, rerupture remains the major risk factor for patients treated non-operatively and this complication has the largest impact on the acute Achilles tendon Total Rupture score (Metz et al., 2011).

Here is some issues that the authors may wish to address:

1. The title is a little confusing. The meta analysis compares of the risk of re-rupture between conservative treatment, open repair surgery, and minimally invasive surgery. The results are based on pairwise comparisons. I would suggest changing the title to make it more informative and easy to the reader.

Authors’ Response: Thanks for your suggestion. We have changed the sentence.

Change to Text: Title page, Page 1, line 1-3: “Rerupture outcome of Conservative Versus Open Repair Versus Minimally Invasive Repair of Acute Achilles Tendon Ruptures: A Systematic Review and Meta-analysis”

2. Abstract Line 39. In the Objective section the authors state that they wanted "to compare the relative effects of rerupture rate...". Is the "relative effect" of re-rupture rate after a certain treatment really addressed in this paper? I suggest changing the first sentence in line 39-41 to simply: To compare the rerupture rate...

Authors’ Response: We have deleted the "relative effect".

Change to Text: Abstract section, Page 3, line 42-44: “To compare the rerupture rate after conservative treatment, open repair, and minimally invasive surgery management of acute Achilles tendon ruptures.”

3. Abstract Line 46: Articles publish before 2010 was excluded. I suggest changing the word "inception" to "2010". Could the authors also explain why this year was chosen for including studies in the meta analysis? The first RCT comparing conservative treatment with surgery for ATR was published in 1981 by Lars Nistor. Since then, many RCTs comparing different treatments have been published. However, very few studies have compared the results after CT, OR, and MIS (3-armed studies). In the literature, there are very few publications of comparative 3-armed studies regarding treatment for ATR before 2010 (Schroeder et al., Treatment of acute Achilles Tendon ruptures: open vs. percutaneous repair vs. conservative treatment: a prospective randomized study. Orthop trans 1997;21:1228. And Majewski et al., Achilles Tendon rupture: a prospective study assessing various treatment possibilities. Orthopade 2000;29:670-6). Some readers might miss those studies in the analyses... Your choice. 

Authors’ Response: We cannot change the word 2010, because some articles have an unclear publication year. Our search strategy included all years of study publication, and two authors independently checked the year of publication through assessing the full-text to exclude articles published before 2010. This process is shown in Table 1. 

There were some reasons why we chose 2010 for including studies in the meta-analysis. The studies published after 2010 have a lower incidence of rerupture (57/1465, 3.89%) than earlier studies (43/500, 8.6%). The reason for the higher rerupture rate may explain by the development of surgical techniques and rehabilitation protocols in the last decade. Since 2010, many emerging technologies emerged that were acceptable, such as the Percutaneous Achilles Repair System (PARS, Arthrex, Inc, Naples, FL). Additionally, after 2010, there were several studies demonstrated that rehabilitation protocols may decline the rerupture rate. We rewrote the sentence, making it more intelligible to readers why this year was chosen for including studies in the meta-analysis. 

It is high quality of comparative 3-armed studies that Schroeder et al. and Majewski et al. performed, however, we decided to include studies that published after 2010 in the final quantitative assessments.

Change to Text: Introduction section, Page 6, line 95-101: “Yet the majority of patients included in these meta-analyses were treated before 2010, with earlier generation devices. These meta-analyses may have represented earlier experience with Achilles tendon ruptures treatment. Since 2010, the percutaneous Achilles Repair System and several minimally invasive techniques have been described[1]. The development of surgical techniques and rehabilitation protocols in the last decade may contribute to lower odds of rerupture[2, 3].”

4. Abstract Line 51: I would suggest using "confidence intervals" instead of "credible intervals".

Authors’ Response: Thanks for your suggestion. We have replaced the word "credible intervals" by "confidence intervals".

Change to Text: Abstract section, Page 3, line 52-54: “Bayesian network meta-analysis with random effects was used to assess pooled relative risks (RRs) and 95% confidence intervals.”

5. Abstract Line 53: Suggest never to start a sentence with a number. Spell 13 out (Thirteen...).

Authors’ Response: We were really sorry for our careless mistakes. Thank you for your reminder. We have spelled ‘13’ out at the beginning of sentence.

Change to Text: Abstract section, Page 4, line 56: “Thirteen trials with a total of 1465 patients were included.”

6. Abstract Line 53-58: The information given here could be re-written and compressed giving the same information with fewer words.

Authors’ Response: We are grateful for the suggestion. To be more clearer and in accordance with the reviewer concerns, we have added a brief description as follows:

Change to Text: Abstract section, Page 4, line 56-61: “Thirteen trials with 1465 patients were included. In direct comparison, there was no difference between open repair and minimally invasive surgery for rerupture rate (RR, 0.72, 95% CI 0.10-4.4; I2=0%; table 2). Compared to the conservative treatment, the RR was 0.27 (95% CI 0.10-0.62, I2=0%) for open repair and 0.14 (95% CI 0.01-0.88, I2=0%) for minimally invasive surgery. The network meta-analysis had obtained the similar results as the direct comparison.”

7. Abstract Line 62: Please state what difference you refer to (re-rupture).

Authors’ Response: Sorry to descript sentence ambiguously. We have rewritten the sentence clearly through adding ‘in rerupture rate’ before ‘between open repair and minimally invasive surgery’.

Change to Text: Abstract section, Page 4, line 68-71: “Both open repair and minimally invasive surgery were associated with a significant reduction in rerupture rate independently compared with conservative management, but no difference in rerupture rate was found comparing open repair and minimally invasive surgery.”

8. Introduction Line 68: Some studies report higher incidences than 31/100.000. Houshian et al. (1996) found an incidence of 37/100.000, and Huttunen et al. found an incidence close to 50/100.000 (Huttunen TT, Kannus P, Rolf C, Felländer-Tsai L, Mattila VM. Acute achilles tendon ruptures: incidence of injury and surgery in Sweden between 2001 and 2012. Am J Sports Med. 2014 Oct;42(10):2419-23. doi: 10.1177/0363546514540599. Epub 2014 Jul 23. PMID: 25056989.)

Authors’ Response: We have quoted these studies that reported higher incidences.

Change to Text: Introduction section, Page 5, line 76-78: “Although Achilles tendon is the strongest and thickest tendon, it is one of the most common tendon ruptures with an annual incidence of 37 to 50 per 100 000 persons, with the largest increase occurring in the middle-aged people[4-6]”

9. Line 71-72: When claiming there are "two types of surgical repair" the reader might wonder if the authors have forgotten about the percutaneous technique (described in ref. 4 and 5). Suggest changing the sentence a bit to avoid this potensial confusion.

Authors’ Response: We have added more information to avoid reader confusion.

Change to Text: Introduction section, Page 5, line 79-81: “Currently available treatments for Achilles tendon ruptures include conservative treatment and two types of surgical repair, open repair and minimally invasive surgery with the percutaneous and mini-open techniques”

10. Line 72-73: Suggest changing focus a bit. For Achilles´ tendon rupture: isn't it the patient who choses his/her treatment based on information given by the physician/surgeon of risks of complications and expected treatment results? And not the opposite? Suggest "The risk of rerupture is a major concern in the shared decision making process (between patient and physician)" or something like that.

Authors’ Response: We really appreciate your comments. It is definite that patients choose the treatment based on information given by the physician/surgeon. We have rewritten this sentence to be more precise.

Change to Text: Introduction section, Page 5, line 84-86: “The risk of rerupture has been a major concern in the shared decision making process between patient and physician.”

11. Line 74: Again: It is not the surgeon, but the patient who choses the treatment, agree?. Furthermore, the trend in treatment of ATR is different between regions of the world: In Asia, surgery is more common than in Scandinavian countries for example. Remember that the literature is shared world wide. Despite that a large number of surgeons in Asia recommend surgical treatment to their patients, there are very few who do so in Scandinavia for instance.

(Yamaguchi S, Kimura S, Akagi R, Yoshimura K, Kawasaki Y, Shiko Y, Sasho T, Ohtori S. Increase in Achilles Tendon Rupture Surgery in Japan: Results From a Nationwide Health Care Database. Orthop J Sports Med. 2021 Oct 21;9(10):23259671211034128. doi: 10.1177/23259671211034128. PMID: 34708136; PMCID: PMC8543583).

(Leino O, Keskinen H, Laaksonen I, Mäkelä K, Löyttyniemi E, Ekman E. Incidence and Treatment Trends of Achilles Tendon Ruptures in Finland: A Nationwide Study. Orthop J Sports Med. 2022 Nov 8;10(11):23259671221131536. doi: 10.1177/23259671221131536. PMID: 36389616; PMCID: PMC9647260.)

(Wilder JH, Ofa SA, Lee OC, Gadinsky NE, Rodriguez RF, Sherman WF. Rates of Operative Management for Achilles Tendon Rupture Over the Last Decade and the Influence of Gender and Age. Foot Ankle Spec. 2022 Jun 13:19386400221102745. doi: 10.1177/19386400221102745. Epub ahead of print. PMID: 35695472.)

Authors’ Response: Again, thanks for bringing this to our attention. Besides, we do appreciate the references. We agree with the view that it is patients who choose the treatment based on information given by the physician/surgeon. We have rewritten the sentence.

Change to Text: Introduction section, Page 5, line 84-86: “The risk of rerupture has been a major concern in the shared decision making process between patient and physician.” 

12. Line 82-83: Regarding change in the risk of re-rupture over time (before or after 2010). It has been claimed that accelerated rehabilitation lowers the risk of re-rupture after conservative treatment. Twaddle & Poon (Early motion for Achilles Tendon ruptures: is surgery important? A randomized prospective study.) introduced this theory in 2007 and was later supported by Soroceanu (Surgical versus non-surgical treatment of acute Achilles tendon rupture: a metaanalysis of randomized trials. J Bone Joint Surg Am 2012;94:2136-43). But it was a secondary finding and therefore only speculative. Later, two prospective cohort studies by Hutchison and Aujla investigated specific accelerated rehabilitation protocols in non-controlled cohorts confirming the low risk of re-rupture. However, the studies were suffering from an insufficient follow up leaving a high risk of patients with rerupture not being registered in the cohorts.

Suggest adding a reference in Line 83 and/or remove "..and rehabilitation protocols.."

(Hutchison AM, Topliss C, Beard D, Evans RM, Williams P. The treatment of a rupture of the Achilles tendon using a dedicated management programme. Bone Joint J. 2015 Apr;97-B(4):510-5. doi: 10.1302/0301-620X.97B4.35314. PMID: 25820890.)

(Aujla RS, Patel S, Jones A, Bhatia M. Non-operative functional treatment for acute Achilles tendon ruptures: The Leicester Achilles Management Protocol (LAMP). Injury. 2019 Apr;50(4):995-999. doi: 10.1016/j.injury.2019.03.007. Epub 2019 Mar 11. PMID: 30898390.)

Authors’ Response: As suggested by the reviewer, we have added more references to support this idea.

Change to Text: Introduction section, Page 6, line 99-101: “The development of surgical techniques and rehabilitation protocols in the last decade may contribute to lower odds of rerupture[2, 3]”

13. Line 87: Suggest removing "relative effects of".

Authors’ Response: Thanks for your suggestion. We have removed "relative effects of".

Change to Text: Introduction section, Page 6, line 105-107: “We carried out this network meta-analysis to compare the rerupture rate after conservative treatment, open repair, and minimally invasive surgery of acute Achilles tendon ruptures.”

14. Line 106-108: The authors claim that the interventions were OR and MIS compared to CT. But in the Results section they report pairwise comparisons. Suggest pairwise comparisons also in the Methods section.

Authors’ Response: We sincerely appreciate the reviewer for careful reading. As suggested by the reviewer, we have added pairwise comparisons also in the Methods section.

Change to Text: Data synthesis section, Page 8-9, line 152-157: “For pairwise meta-analysis, we used a random effects model to compute pooled effect sizes, and risk ratio for outcomes with 95% confidence intervals.

For network meta-analysis, we used a random consistency model to compute the study effect sizes, and binomial likelihood arguments for the rerupture outcome”

15. Line 114: Change "will be" to "were".

Authors’ Response: Thanks for your suggestion. We have changed "will be" to "were".

Change to Text: Study selection section, Page 8, line 134-135: “Discrepancies between reviewers were resolved by discussion or consulting a third author (YZ).”

16. Line 132-133: Grammatically error.

Authors’ Response: Thanks for your suggestion. We have checked the sentence, and corrected it.

Change to Text: Data synthesis section, Page 9, line 154-155: “For network meta-analysis, we used a random consistency model to compute the study effect sizes, and binomial likelihood arguments for the rerupture outcome.”

17. Line 142: Remove "was".

Authors’ Response: Thanks for your suggestion. We have removed "was".

Change to Text: Data synthesis section, Page 9, line 166: “P-value less than 0.05 suggests the consistency of the model is satisfactory.”

18. Line 163-173: This section regarding reruptures can be shortened without losing information.

Authors’ Response: Thanks for your suggestion. We have shortened this section

Change to Text: Results Rerupture section, Page 11, line 190-202: “The pair-wise meta-analysis pooled effects showed that no difference between open repair and minimally invasive surgery for rerupture rate (RR, 0.72, 95% CI 0.10-4.4; I2=0%; table 2). Compared to the conservative treatment, the RR was 0.27 (95% CI 0.10-0.62, I2=0%) for open repair and 0.14 (95% CI 0.01-0.88, I2=0%) for minimally invasive surgery. The network meta-analysis had obtained the similar results as the direct comparison (Supplemental Figure 1). No statistically significant differences were found in rerupture rate between open repair and minimally invasive surgery.”

19. Line 184: Please be consistent. Chose "non-operative" or "conservative" throughout the paper. Also specify both OR and MIS.

Authors’ Response: Thanks for your valuable suggestion. We have checked the abbreviation mistake, and made it consistent throughout the whole manuscript.

Change to Text: Discussion section, Page 12, line 219-221: “This systematic review and network meta-analysis of RCTs performed a comparison among conservative treatment versus open repair versus minimally invasive surgery for acute Achilles tendon ruptures.”

20. Line 187: Same as point 7.

Authors’ Response: Thanks for your suggestion. We have rewritten the sentence clearly.

Change to Text: Discussion section, Page 12, line 224-225: “..., but no difference in rerupture rate was found comparing open repair and minimally invasive surgery.”

21. Line 189: operative treatment (open repair and minimally invasive surgery)

Authors’ Response: Thanks for your valuable suggestion. We have added explanatory note after “operative treatment”.

Change to Text: Discussion section, Page 13, line 228-230: “The previous study has demonstrated that operative treatment (open repair and minimally invasive surgery) of acute Achilles tendon ruptures could reduce the risk of re-rupture compared with nonoperative treatment.”

22. Line 201: Change "them" to "studies"

Authors’ Response: Thanks for your suggestion. We have changed "them" to "studies"

Change to Text: Discussion section, Page 13, line 242-243: “Both studies compared the rerupture rate between open repair and minimally invasive surgery of acute Achilles tendon ruptures.”

23. Line 202: Use either abbreviations on both open repair and minimally invasive surgery, or spell out for both, and be consistent throughout the whole manuscript.

Authors’ Response: Thanks for your valuable suggestion. We have checked the abbreviation mistake, and made it consistent throughout the whole manuscript.

Change to Text: Discussion section, Page 13, line 242-243: “Both studies compared the rerupture rate between open repair and minimally invasive surgery of acute Achilles tendon ruptures.”

24. Line 203-204. The language can be shortened throughout the manuscript. Here is an example of how the information could be presented: "The pooled effect showed no relevant differences in re-rupture rate between the two techniques."

Authors’ Response: Thanks for your valuable suggestion. This example was a concise presentation, we have shortened the sentence to “The pooled effect showed no relevant differences in re-rupture rate between the two techniques.” 

Change to Text: Discussion section, Page 13, line 243-244: “The pooled effect showed no relevant differences in re-rupture rate between the two techniques.”

25. Line 207: abbreviations or spell out.

Authors’ Response: Thanks for your valuable suggestion. We, have spelled out “MIS”.

Change to Text: Discussion section, Page 14, line 249: “…, and minimally invasive surgery for acute Achilles tendon ruptures.”

26. Line 223-225: The conclusion is well written, but consider removing "independently" in line 224 and state "no difference in rerupture rate was found comparing open repair and minimally invasive surgery".

Authors’ Response: Thanks for your valuable suggestion. We have removed " independently, and stated "no difference in rerupture rate was found comparing open repair and minimally invasive surgery".

Change to Text: Conclusion section, Page 15, line 270-273: “The results of the present study suggest that both open repair and minimally invasive surgery were associated with a significant reduction in rerupture rate compared with conservative management, but no difference in rerupture rate was found comparing open repair and minimally invasive surgery.”

27. Line 403: Table 1 is Characteristics of included randomized controlled trials (spell out). Not Baseline characteristics.

Authors’ Response: Thanks for your valuable suggestion. We, have changed “Baseline characteristics” to “Characteristics of included randomized controlled trials”

Change to Text: Results section, Page 10, line 182-184: The characteristics of included randomized controlled trials were displayed in Table 1.

Page 22, line 458: “Characteristics of included randomized controlled trials.”

28. Table 1: In the last study (Myhrvold 2022): add (WB at 2 wk) also after non-operative treatment.

Authors’ Response: thanks for your valuable suggestion. We, have added “(WB at 2 wk)” in the last study (Myhrvold 2022) in Table 1

Change to Text: Page 23 (part of table 1): 

Myhrvold 2022 178 176 172 42/136 44/132 49/123 39.9 (8.1);

39.9 (8.9);

39.1 (8.4) 12 plaster cast for 2 wk + orthosis for 6 wk (WB at 2 wk);

OR + plaster cast for 2 wk + orthosis for 6 wk (WB at 2 wk); 

MIS + plaster cast for 2 wk + orthosis for 6 wk (WB at 2 wk)

* Data are shown as median (range)

WK: week; OR: open repair; MIS: minimally invasive surgery; WB: weightbearing

29. Table 2 Line 406-412: Please do not use abbreviations in table and figure descriptions. Spell out. In my opinion table 2 is to difficult to grasp for the reader. Consider to present the pairwise comparisons more clearly with three columns but with the upper row stating Conservative treatment vs open repair; Conservative treatment vs minimally invasive surgery; Open repair vs minimally invasive surgery.

Authors’ Response: Thanks for your valuable suggestion. We, have spelled all abbreviations out in table and figure descriptions. Re-configuration of table 2 as follow:

Change to Text: Page 24:

Table 2. Comparison of different treatment in AAT between pair-wise meta-analysis and network meta-analysis.

 Conservative treatment vs open repair Conservative treatment vs minimally invasive surgery Open repair vs minimally invasive surgery

pair-wise meta-analysis 0.27 (0.11, 0.59) 0.17 (0.04, 0.71) 0.62 (0.14, 2.63)

network meta-analysis 0.27 (0.10, 0.62) 0.14 (0.01, 0.88) 0.72 (0.10,4.4)

Figure 3: Node-splitting analysis of inconsistency for rerupture rate

In conclusion I find the result of this study very interesting and important. In my opinion, however, prior to publication, the manuscript needs to be revised. I wish the authors well in further preparation.

Reviewer #3: The authors present an interesting study comparing the relative effects between conservative, open, and minimally invasive Achilles tendon repair for Achilles tendon rupture. They demonstrated that no differences exist in re-rupture rate between the open and minimally invasive techniques; however, but of these techniques had less re-rupture rates compared to conservative treatment.

Several questions arise regarding this study. What was the length of time post-operative that these patients were followed. Several studies have shown that operative management of Achilles tendon rupture have improvement in early re-rupture, but what about long term re-rupture rates between these techniques? (PMID: 7551757) 

Authors’ Response: thanks for your valuable suggestion. The length of time post-operative that the most studies we included was about 12 to 24 months, and we have added this description in results section. Given the 12 to 24 months length of time post-operative that these majority patients were followed, we have no appropriate date to analyze the outcome of long term rerupture rate. We have added the outcome of the long term rerupture rate to limitation section in this paper. Additionally, we conducted a sensitive analysis to excluded the influence of the study that had 188 months follow-up. Through analysis, we obtained a similar result.

Change to Text: Results section, Page 10, line 178-179: The length of time post-operative that the most studies we included was about 12 to 24 months.

Limitations section, Page 14, line 259-263: “We failed to obtain appropriate date to analyze the outcome of long-term rerupture rate due to those majority patients were followed up to 12 to 24 months. Additionally, we conducted a sensitive analysis to excluded the influence of the study that had a 188 months follow-up. Through analysis, we obtained a similar result.”

Unfortunately, as stated by the authors, several systematic reviews comparing these operative techniques have been performed and at a much more rigorous level. I fail to see how this study contributes anything new to the field of Achilles tendon rupture. What separates this study from those other studies? Why were other complications like sural nerve injury, wound dehiscence, VTE, surgical site infections not examined? Inclusion of these data would greatly improve your data.

Authors’ Response: Thanks for your valuable suggestion. We have tried best to make novel idea in this paper. The present study has several novelty. Firstly, we included the most large RCT published in 2022, which accounting for 40% (526/1316) of the total number of patients in the last study (Meulenkamp et al., 2021). This produces more robust estimates than previous meta-analyses. Secondly, we excluded studies published before 2010 to eliminate the impact of early techniques and rehabilitation protocols. The studies published after 2010 have a lower incidence of rerupture (57/1465, 3.89%) than earlier studies (43/500, 8.6%). Thirdly, we adopted direct and indirect method to compare the rerupture rate between these techniques.

According to viewers valuable suggestion, we have additionally included other complications to improve our data, including infections, DVT, nerve injury to improve our data. We failed to examine Pulmonary embolism due to only two studies report it in this paper. We had added this into limitation section.

Change to Text: 

Eligibility criteria section, Page 7, line 128-129: Secondary outcomes included wound infection, sural nerve injury, and deep vein thrombosis.

Results section, Page 11-12, line 204-208:

Other outcomes

Compared to conservative treatment, open repair management had significant higher infection rate, however, with very wide confidence intervals (Supplemental Figure 3). There was a significant difference between conservative treatment and open repair management in deep vein thrombosis complication (Supplemental Figure 4). We have not found any significant difference in sural nerve injury (Supplemental Figure 5).

Limitations section, Page 14, line 263-264: “We failed to examine the complication of Pulmonary embolism due to only two studies report it in this paper.”

Supplemental Figure 3: Node-splitting analysis of inconsistency for the wound infection.

 

Supplemental Figure 4: Node-splitting analysis of inconsistency for the deep vein thrombosis.

 

Supplemental Figure 5: Node-splitting analysis of inconsistency for the sural nerve injury.

 

The configuration of Table 2, which is the crux of your data, is extremely difficult to interpret. Re-configuration of this data would greatly improve the main message of this study.

Authors’ Response: thanks for your valuable suggestion. We, have Re-configurated of this data 

Change to Text: Page 24:

Table 2. Comparison of different treatment in AAT between pair-wise meta-analysis and network meta-analysis.

 Conservative treatment vs open repair Conservative treatment vs minimally invasive surgery Open repair vs minimally invasive surgery

pair-wise meta-analysis 0.27 (0.11, 0.59) 0.17 (0.04, 0.71) 0.62 (0.14, 2.63)

network meta-analysis 0.27 (0.10, 0.62) 0.14 (0.01, 0.88) 0.72 (0.10,4.4)

As the main analysis was performed by XC and HD with YZ as the tie breaker, further explanation of the credentials of these reviewers is paramount to state to the reader. What qualifies these reviewers to understand, analyze, and interpret all this data?

Authors’ Response: HD is an orthopedist, XC is a resident, and YZ is an evidence-based medical expert. We have added the information in Title page 

Change to Text: Title page, Page 1:

“Haidong Deng MD1, Xin Cheng MD4, Yi Yang MD1, Fang Fang MD2, Jialing He MD2, Yixin Tian MD2, Tiangui Li MD5, Yangchun Xiao MD3, Yuning Feng MD1, Peng Wang MD2, Weelic Chong MD-PhD candidate6, Yang Hai MD6, Yu Zhang MD1,2

Affiliations

1Department of Orthopedic, 2Center for Evidence Based Medical, 3Department of Neurosurgery, Affiliated Hospital of Chengdu University, Chengdu, Sichuan, China

4 West China Hospital, Sichuan University, Chengdu, Sichuan, China 

5The First People's Hospital of Longquanyi District Chengdu, Sichuan, China

6 Thomas Jefferson University, Philadelphia, PA”

This was a great effort by the authors to help determine best course of action for a growing problem especially in the post-COVID era where Achilles tendon ruptures are rampant as people are slowly returning to their normal athletic activities after several years without normal athletic activity patterns. Answering the above questions will greatly assist in the publication of this study. 

Authors’ Response: Thanks for your valuable suggestion.  

Reference:

1. Patel MS, Kadakia AR. Minimally Invasive Treatments of Acute Achilles Tendon Ruptures. Foot Ankle Clin. 2019;24(3):399-424. Epub 2019/08/03. doi: 10.1016/j.fcl.2019.05.002. PubMed PMID: 31370993.

2. Hutchison AM, Topliss C, Beard D, Evans RM, Williams P. The treatment of a rupture of the Achilles tendon using a dedicated management programme. Bone Joint J. 2015;97-b(4):510-5. Epub 2015/03/31. doi: 10.1302/0301-620x.97b4.35314. PubMed PMID: 25820890.

3. Aujla RS, Patel S, Jones A, Bhatia M. Non-operative functional treatment for acute Achilles tendon ruptures: The Leicester Achilles Management Protocol (LAMP). Injury. 2019;50(4):995-9. Epub 2019/03/23. doi: 10.1016/j.injury.2019.03.007. PubMed PMID: 30898390.

4. Huttunen TT, Kannus P, Rolf C, Felländer-Tsai L, Mattila VM. Acute achilles tendon ruptures: incidence of injury and surgery in Sweden between 2001 and 2012. Am J Sports Med. 2014;42(10):2419-23. Epub 2014/07/25. doi: 10.1177/0363546514540599. PubMed PMID: 25056989.

5. Houshian S, Tscherning T, Riegels-Nielsen P. The epidemiology of Achilles tendon rupture in a Danish county. Injury. 1998;29(9):651-4. Epub 1999/04/22. doi: 10.1016/s0020-1383(98)00147-8. PubMed PMID: 10211195.

6. Egger AC, Berkowitz MJ. Achilles tendon injuries. Curr Rev Musculoskelet Med. 2017;10(1):72-80. Epub 2017/02/15. doi: 10.1007/s12178-017-9386-7. PubMed PMID: 28194638; PubMed Central PMCID: PMCPMC5344857.

---

## [Editor Report · Decision Letter 1]

14 Apr 2023

Rerupture outcome of Conservative Versus Open repair Versus Minimally Invasive Repair of Acute Achilles Tendon Ruptures: A Systematic Review and Meta-analysis

PONE-D-22-33292R1

Dear Dr.Zhang

We’re pleased to inform you that your manuscript has been judged scientifically suitable for publication and will be formally accepted for publication once it meets all outstanding technical requirements.

Kind regards,

Daniel Ramskov, Ph.D

Academic Editor

PLOS ONE
---

## [Editor Report · Acceptance letter]

24 Apr 2023

PONE-D-22-33292R1 

Rerupture outcome of Conservative Versus Open repair Versus Minimally Invasive Repair of Acute Achilles Tendon Ruptures: A Systematic Review and Meta-analysis 

Dear Dr. Zhang:

I'm pleased to inform you that your manuscript has been deemed suitable for publication in PLOS ONE. Congratulations! Your manuscript is now with our production department. 

Kind regards, 

on behalf of

Dr. Daniel Ramskov 

Academic Editor

PLOS ONE